# Pain Perception in Disorder of Consciousness: A Scoping Review on Current Knowledge, Clinical Applications, and Future Perspective

**DOI:** 10.3390/brainsci11050665

**Published:** 2021-05-20

**Authors:** Rocco Salvatore Calabrò, Loris Pignolo, Claudia Müller-Eising, Antonino Naro

**Affiliations:** 1IRCCS Centro Neurolesi Bonino Pulejo, 98123 Messina, Italy; 2Istituto Sant’Anna, 88900 Crotone, Italy; l.pignolo@gmail.com; 3Center of Neurorehabilitation Neuroneum, 61348 Bad Homburg, Germany; claudia.mueller-eising@neuroneum.de; 4Department of Clinical and Experimental Medicine, University of Messina, 98124 Messina, Italy; g.naro11@alice.it

**Keywords:** prolonged disorders of consciousness (PDOC), unresponsive wakefulness syndrome (UWS), minimally conscious state (MCS), pain, nociception, functional connectivity, neurophysiology

## Abstract

Pain perception in individuals with prolonged disorders of consciousness (PDOC) is still a matter of debate. Advanced neuroimaging studies suggest some cortical activations even in patients with unresponsive wakefulness syndrome (UWS) compared to those with a minimally conscious state (MCS). Therefore, pain perception has to be considered even in individuals with UWS. However, advanced neuroimaging assessment can be challenging to conduct, and its findings are sometimes difficult to be interpreted. Conversely, multichannel electroencephalography (EEG) and laser-evoked potentials (LEPs) can be carried out quickly and are more adaptable to the clinical needs. In this scoping review, we dealt with the neurophysiological basis underpinning pain in PDOC, pointing out how pain perception assessment in these individuals might help in reducing the misdiagnosis rate. The available literature data suggest that patients with UWS show a more severe functional connectivity breakdown among the pain-related brain areas compared to individuals in MCS, pointing out that pain perception increases with the level of consciousness. However, there are noteworthy exceptions, because some UWS patients show pain-related cortical activations that partially overlap those observed in MCS individuals. This suggests that some patients with UWS may have residual brain functional connectivity supporting the somatosensory, affective, and cognitive aspects of pain processing (i.e., a conscious experience of the unpleasantness of pain), rather than only being able to show autonomic responses to potentially harmful stimuli. Therefore, the significance of the neurophysiological approach to pain perception in PDOC seems to be clear, and despite some methodological caveats (including intensity of stimulation, multimodal paradigms, and active vs. passive stimulation protocols), remain to be solved. To summarize, an accurate clinical and neurophysiological assessment should always be performed for a better understanding of pain perception neurophysiological underpinnings, a more precise differential diagnosis at the level of individual cases as well as group comparisons, and patient-tailored management.

## 1. Introduction

Prolonged disorders of consciousness (i.e., more than 28 days) (PDOC) represent a possible evolution of coma, a condition following severe brain injury, characterized by a loss of wakefulness and awareness (mainly owing to a lesion in the ascending reticular system or diffuse (bi)hemispheric damage) [1,2]. PDOC mainly comprises unresponsive wakefulness syndrome (UWS, namely, vegetative state) and the minimally conscious state (MCS) [3,4]. The former entity is characterized by three (all necessary) clinical criteria: (i) no evidence of awareness of self or environment at any time; (ii) no response to any stimuli suggesting purposeful behavior; and (iii) no evidence of language comprehension or expression. Additionally, cycles of eye closure and eye-opening are preserved, giving the appearance of a sleep–wake cycle. Lastly, hypothalamic and brainstem function are preserved [2,3,4]. The absence of response to commands or voluntary oriented movements in the presence of wakefulness is thus the main feature of UWS. The latter entity is a condition of severely altered consciousness in which minimal but definite behavioral evidence of self or environmental awareness is appreciable. Therefore, MCS patients show inconsistent but reproducible or sustained and cognitively mediated behaviors associated with conscious awareness, differently from reflexive behavior as compared to UWS patients [5].

Consistently with this premise, UWS patients should not perceive anything, including pain, even if they can show reflexive response to stimuli, including the nociceptive, but without any stimulus-related differentiation (i.e., purely reflexive behaviors). The discovery of residual traces of awareness, covert awareness, and brain activity modulation in response to command and stimuli in some patients clinically labeled as UWS using advanced neuroimaging or electrophysiological techniques put in crisis the former assumption that patients with UWS are unable to perceive anything, including pain. Some individuals may have a dissociation of motor and cognitive function [6], i.e., they may retain some focal brain activations somehow sustaining conscious perception, despite the absence of integrated large-network processes known to sustain consciousness [7,8,9,10,11,12,13,14,15,16]. This points to a somehow preserved awareness in the presence of reflexive motor output [5,17,18,19,20]. Behavioral responsiveness in PDOC patients appears to be mediated by three-order network interactions [9,10,15,16]: (i) disconnection between lower-level cortices (preserved activation) and higher-order associative cortices (impaired activation) [21]; (ii) preserved activation of higher-order associative cortices [22,23,24]; and (iii) consciousness recovery related to the restoration of thalamo–cortical connectivity [25,26].

It remains, therefore, unclear whether and how UWS patients can show signs of possible covert capacity for subjective pain perception, potentially indicating that some patients behaviorally diagnosed as with UWS may have a residual capacity of pain perception (and are thus in MCS). Nonetheless, we may argue that there are good reasons to think that these patients could experience pain. This argument stems from three main issues.

Firstly, the efficacy of clinical assessments on detecting aware pain responses is limited. Pain assessment in PDOC individuals is limited to the clinical appreciation of responses to nociceptive stimuli using, e.g., the Coma Recovery Scale-Revised (CRS-R) [27] or the Nociception Coma Scale-Revised (NCS-R) [28,29,30,31]. The responses to nociceptive stimuli include stereotyped responses (i.e., slow generalized flexion or extension of the upper and lower extremities), flexion withdrawal (i.e., withdrawal of the limb away from the point of the stimulation), and localization responses (i.e., the non-stimulated limb locates and makes contact with the stimulated body part at the point of stimulation), all of which are linked to brainstem, subcortical, or cortical activity, respectively (nociceptive network or pain matrix) [32,33,34,35]. However, the magnitude, consistency, and repeatability of behavioral responses to nociceptive stimuli are limited by many factors, including paralysis and spasticity. Furthermore, retrospective reports suggest that patients with PDOC may exhibit some form of appraisal of noxious stimuli. Once recovered awareness, some individuals reported vivid memories of pain and other unpleasant/uncomfortable situations (e.g., noise and sleep deprivation) [35]. 

Secondly, the evidence of brain activations within some brain areas concerning nociceptive stimulation in a way resemble those detectable in aware individuals [33,36,37]. More in-depth, several studies propose that brain mechanisms underlying affective consciousness can survive despite very severe lesions that make higher conscious functions such as attention, working memory, or language comprehension impossible [19,36,37,38,39]. Regardless of whether this is due to misdiagnosis or technical limitations, these data contravene the tenet that patients with UWS cannot experience pain. They may not be capable of exhibiting a detectable reaction to painful stimuli, albeit being capable of perceiving them [40]. 

Thirdly, if a patient is aware, they are presumably able to feel pain, because if awareness does not imply sentience, the absence of awareness excludes any sentience. Some works showed isolated cortical activations that document retained modular function without the integrative processes necessary for consciousness [11,36,38,39,41,42,43]. Notably, patients in MCS showed a broad network activation similar to normal controls [36] following nociceptive stimulation. Instead, individuals in UWS also showed some activation in those similar networks, although it was much less prominent and more focal than MCS and control individuals did [36,38]. These observations are consistent with the issue of a brain damage-induced, large-scale connectivity breakdown among the higher-order association cortices and the primary cortical areas [44,45]. However, they challenge the assumption that patients in a UWS cannot experience and report a painful experience, given that some cortical activations are appreciable. In this regard, neurophysiological and neuroimaging studies showed brain activations following nociceptive stimulation within the brain structure primarily involved in affective–cognitive pain processing (including the anterior insula, the ACC, and the prefrontal cortex), sometimes occurring at multiple levels, even in patients with UWS [46,47,48,49]. These activations go beyond those involved in encoding the sensory-discriminative information (including primary and secondary somatosensory cortices, the lateral thalamus, and the posterior insula) [46,47,48,49] (Figure 1).

These data could suggest that brain activations, when complex, are a marker of ongoing conscious perception or pain awareness [50]. Furthermore, these data could imply that the identification of potential neurophysiological markers of conscious pain perception in PDOC patients may contribute to differentiating between UWS and MCS. Nonetheless, growing evidence suggests that false positive and false negative findings may result from applying different statistical methods to patient data [51]. This statistical bias may challenge the use of these paradigms in the clinical setting, with particular regard to three common clinical scenarios where the risk of diagnostic error may be most pronounced in this patient group, i.e., disclosure of results to patients’ families, withdrawal of life-sustaining therapies, and equitable distribution of medical resources [51,52]. However, contradictory findings may also stem from other issues [53]. Indeed, various sources can explain the wide variety with regard to sensitivity and specificity of electrophysiological techniques in detecting consciousness in patients with DoC. These include task robustness of active paradigms (including personally relevant stimuli, salient self-referential stimuli, the instruction to count relevant stimuli), patient’s features potentially preventing them from responding in active tasks despite being conscious (including perceptual and cognitive impairment, arousal fluctuations, cognitive fatigue), small sample sizes, and methodological issues (including equipment, paradigms, blinding, and artifacts) [53].

Consistently with these issues, studies showing neurophysiological pain signatures in patients with PDOC at an individual level are not conclusive. Therefore, to implement clinical use of neurophysiological techniques in combination with behavioral assessment of pain, one needs to know the sensitivity and specificity of these methods, and which stimuli are the most potent in detecting pain responses.

Another essential issue to consider is that pain may represent a chronic, lasting condition in patients with PDOC, as a sort of inner nature of PDOC itself owing to central sensitization mechanisms leading to pain hypersensitivity [54]. In this regard, it is helpful to remember the distinction between nociception and pain. The former refers to the neural mechanisms of encoding and processing an actually or potentially tissue-damaging event [54]. The latter is “an unpleasant sensory and emotional experience associated with actual or potential tissue damage, or described in terms of such damage”, as per the International Association for the Study of Pain (IASP) definition [55]. Therefore, nociception and pain can occur each without the other (e.g., pain in amputated limbs, fibromyalgia), depending on several biological, cognitive, emotional, social, and behavioral factors. Therefore, more profound knowledge of the effects of patients’ clinical features and comorbidity profile on pain processing is necessary to better understand pain perception.

### Objective

The functional neuroimaging devices used to estimate pain perception in PDOC patients are limitedly available in rehabilitation and neurologic centers, and not all of the patients with PDOC can be subjected to such devices. Conversely, neurophysiological paradigms are more accessible and adaptable to clinical needs. In this scoping review, we sought to investigate the neurophysiological basis underpinning pain perception in PDOC, suggesting that its assessment might help in reducing the PDOC misdiagnosis rate. We adopted a scoping review to describe the landscape of available evidence, identify gaps in the existing literature, and illuminate areas for further research. A scoping review methodology was also selected because of the relatively circumscribed nature of the field.

## 2. Methods

### 2.1. Eligibility Criteria

To be included in the present scoping review, papers needed to deal with or focus on neurophysiologic evaluations of pain perception in PDOC individuals. Peer-reviewed journal papers were included without time restrictions if they described a measure for pain in relation to awareness, involved adult human participants, and were written in English. Studies such as randomized clinical trials, case-studies, and reviews (systematic reviews and meta-analyses) were included in order to consider different aspects of assessing pain in relation to awareness. Papers were excluded if they did not fit into the conceptual framework of the study, focused on a non-PDOC condition (i.e., lasting less than 28 days), or belonged to gray literature, including letters, commentaries, textbook chapters, technical or theoretical descriptions, abstracts, and conference proceedings. Furthermore, it was ascertained that the study patients were provided with a behavioral assessment tool for assessing the level of consciousness, i.e., the CRS-R, which has acceptable standardized administration and scoring procedures, excellent content validity, substantial evidence of good interrater reliability, and it is the only scale to address all Aspen Workgroup criteria [56]. In addition, it was ascertained that pain-related behavioral assessment was conducted using tools with suitable psychometric properties for the assessment and detection of pain in non-communicative severely brain-injured patients, including the NCS-R [28,29,30,31,32,33,42,47,57,58].

### 2.2. Information Sources and Search

We searched the following bibliographic databases to identify the potentially relevant documents: PubMed, MEDLINE, Google Scholar, and Cochrane Database. The search strategies were drafted by L.P. and A.N. and further refined by C.M.E., who also settled all disagreements regarding article inclusion at each review phase. The final search strategy included the keywords vegetative state/UWS OR minimally conscious state AND pain.

### 2.3. Selection of Sources of Evidence

The research returned 261 papers (pruned from duplicates, which were removed by A.N.). The authors L.P., A.N., and R.S.C. discussed the titles and the abstracts, and amended the screening and data extraction manual before beginning screening for this review, to increase data consistency. Thus, 59 full-text articles were identified to be retrieved and assessed for eligibility by using the abovementioned keywords, main judgment criteria, and publication types. Both L.P. and A.N. independently reviewed all articles, sequentially evaluating the titles, abstracts and then the full text of all publications identified by our searches for potentially relevant publications. The author C.M.E. settled all disagreements regarding article inclusion at each review phase discussion with other reviewers if needed.

Of the identified 59 full-text articles, 43 were excluded because they did not directly quantify pain in relation to awareness (*n* = 31), or were not original quantitative research (*n* = 12). The remaining 16 studies were considered eligible for this review (see [59,60,61,62,63,64,65,66,67,68,69,70,71,72,73,74,75,76,77,78,79,80,81,82,83,84,85], including reviews). Figure 2 illustrates the search strategy we used to select pain assessment studies in patients with PDOC. All the reviewed studies are summarized in Table 1.

## 3. Results

Generally, objective measures of pain processing in individuals with PDOC consist of the evaluation of spontaneous responses or behaviors concerning pain (such as eye-opening, variations in breathing, heart rate, and blood pressure, and occasionally grimace-like or crying-like behavior) [50], are elicited using nociceptive stimuli (aimed at highlighting large-scale brain activations potentially related to awareness), or are elicited by active stimulation paradigms (aimed at representing functional communication and willful modulation of brain activity related to aware pain processing) [40,42,46,47,57,58]. In the present review, we grouped the studies by the types of measure they analyzed, and summarized the type of settings, populations, and study designs for each group, along with the measures used and broad findings. The reviewed evidence is presented in a narrative format and with a summary table containing reference, sample, methods, findings, and conclusions (Table 1).

### 3.1. Autonomic Nervous System

These measures include heart rate variability (HRV) and blood pressure, which qualify as independent indicators of the interactions between the central nervous system (CNS) and the autonomic nervous system (ANS) [33,59,60]. The assessment of such two-way interaction is proposed as necessary for PDOC assessment, because it reflects the continuous modulation of homeostatic processes and allostatic adaptation to internal or external necessities [61,62,63]. Notably, an integer CNS–ANS interaction is necessary to modulate the autonomic output in response to pain and emotional, behavioral, or cognitive stimuli [61,64,65,66,67,68]. Therefore, a preserved ANS responsiveness to purposeful stimuli might suggest residual awareness [60,69]. One could concern that these measures are of subcortical origin; thus, they do not necessarily reflect conscious pain perception. However, advanced quantitative assessments of the measures, including circadian regulation of HRV and its correlation with other more complex neurophysiological measures, e.g., laser-evoked potential (LEP) power spectra and EEG connectivity measures [52], highlighted a fine cortico–subcortical regulation of such measures in some patients. This fine cortico–subcortical regulation presupposes partial integrity of distinct thalamo–cortical loops, which are well-known prerequisites for awareness to emerge, even covertly. To support this issue, it has recently been proposed that quantitative features of HRV in response to potentially noxious and non-noxious stimuli were significantly correlated with the CRS-R scoring, despite a reduced ANS tuning in response to noxious and non-noxious stimuli. A less complex ANS response to noxious stimuli characterizes UWS patients and correlates with a behaviorally estimated awareness preservation [70]. Similarly, short- and long-term ANS responsiveness to noxious and non-noxious stimuli may also correlate with behavioral responsiveness and, thus, awareness degree [69].

Another work focused on PDOC patients’ electrodermal activity. The authors found that cutaneous responses were preserved only in the UWS patients who regained consciousness [71,72]. Similarly, some studies identified skin conductance responses (SCRs) of different magnitude in response to varying auditory (including white noise, music, relatives’ voice, and patients’ name) and somatosensory (tactile) stimuli, thus suggesting a preserved ability to discern between stimuli, which presupposes awareness [16,73,74].

### 3.2. Laser-Evoked Potentials and Advanced EEG Signal Analyses

LEPs were identified in all the patients with a significant N2 and P2 latency increase, despite the absence of late somatosensory potentials showing a significant N2 and P2 latency increase, and the presence of auditory mismatch negativity in all the patients [75]. Later, the same group showed that UWS patients presented with a variable degree of preservation of multisensory EPs compared to healthy subjects except for LEPs, which were recognized in all the patients [76]. Differential preservation of EPs has recently been confirmed by recording ANS and LEPs while receiving touch- and pain-related stimulations in PDOC individuals. The authors found significant frontal and parietal activations in response to pain stimuli in the delta frequency range, which mirror the basic attention orientation and perceptual processes. Furthermore, the nociceptive stimulation yielded a more consistent and informative pattern of covert response [77].

Further confirmation and deepening of the matter came from some studies illustrating that LEPs were with lower amplitudes and more delayed latencies in UWS individuals than in MCS persons [78]. This discrepancy suggests the possibility to differentiate between UWS and MCS, but it does not infer awareness preservation. To this end, it was attempted to discern between Aδ-LEP and C-LEP in PDOC patients [79]. Given that some patients in UWS showed only the C-LEP, it was hypothesized that the cortical generators of these components are more likely to survive a severe brain injury and may represent a valuable tool for instrumental pain assessment in the most damaged patients. However, these data do not add to awareness preservation. Conversely, a combination of motor-evoked potentials (MEPs) and LEPs to investigate pain–motor integration (PMI) in post-anoxic UWS patients outlined no significant differences in the resting motor threshold between UWS patients and healthy subjects, and a significantly compromised PMI in UWS patients as compared to healthy subjects with some patients showing signs of partially restored PMI [80]. Thus, PMI was considered as a marker of covert awareness because its sensitivity to non-invasive neuromodulation was differently preserved in behavioral and non-behavioral individuals.

Additional data on residual awareness preservation came from studies that deeply analyzed LEP features. In one study, the effects of repetitive laser stimulation on gamma-band oscillation (GBO) power were evaluated [81]. An increase in GBO power and NCS-R score in MCS patients was observed, whereas there was not a significant increase in GBO power and NCS-R score in the UWS group except for in five patients. In line with these data [82], it was estimated that there was a correlation between LEP and 24 h polysomnography data in PDOC individuals. The authors found that higher LEP latencies and lower LEP amplitudes were appreciable in patients with UWS compared to those with MCS and that this difference was correlated with a more preserved sleep–wake cycle and sleep structure in patients with MCS as compared to those with UWS.

Subsequent studies analyzed the spectral features of C-LEP and confirmed that a fine cortico–subcortical regulation of such measures exists in some UWS patients. This presupposes partial integrity of distinct thalamo–cortical loops, which are a well-known prerequisite for awareness to emerge, even covertly. Furthermore, when investigating the inter-peak interval (IPI) between the N2 and P2 components of LEP, a correlation between IPI and the NCS-R was found, suggesting that IPI could represent an objective marker of behavioral responsiveness to nociceptive stimulation independent from the sensory part of pain processing, which is critically influenced by the stimulation intensity [83]. In this regard, partially preserved γ-band LORETA activation and ERP γ-power magnitude induced by laser stimulation was found in MCS and some UWS individuals. Notably, two individuals showed strong limbic activation, signifying that some UWS patients may somehow perceive the affective components of nociceptive stimulation beyond pain sensory processing preservation [84]. The preservation of large-scale networks related to cognitive pain processing in some UWS individuals has also been suggested using non-invasive brain stimulation. This approach was applied to the anterior cingulate cortex to modulate the GBO in the centroparietal areas (considered as a marker of either subjective pain perception processes or pain-related motor behavior preparation) and the latency and amplitude of cortical nociceptive potentials. A significant modulation was appreciable in two UWS individuals, suggestive of a conscious pain processing even in patients with severe PDOC [85].

Distinguishing between studies that showed neurophysiological pain signatures in patients with PDOC on an individual-level beyond a group level is crucial if one aims to reason for clinical utility of these methods. Actually, in a clinical setting, one needs to know the sensitivity and specificity of these techniques in detecting subjective pain perception (i.e., at an individual level). We did not systematically investigate these aspects, but the studies we reviewed did not identify significant within-group differences concerning the neurophysiological pain signature, with very few exceptions concerning a few patients that were tested using different, advanced methodological approaches (including GBO, C-LEP spectra, and pain–motor integration) [16,73,74].

Furthermore, neurophysiological pain evaluation may play a role in the prognosis of awareness recovery. In this regard, a 68-year-old woman with a five-year UWS secondary to a severe brain hemorrhage was prognosticated as likely suffering from a cognitive–motor dissociation syndrome three years before using a neurophysiological protocol based on a PMI assessment. This protocol shows extensive nociceptive processing within large frontoparietal networks. In fact, after three years, the subject emerged from UWS and then from MCS (as being able to communicate appropriately) [86].

## 4. Discussion

Pain diagnosis and management in PDOC is still a thorny issue. The potential to experience pain and suffering is frequently raised concerning treatment, ethical, and legal questions in individuals with PDOC. To date, the international practice guidelines for DPC [87] recommend (no. 13) that “Clinicians should assess individuals with a PDOC for evidence of pain or suffering and should treat when there is reasonable cause to suspect that the patient is experiencing pain (Level B), regardless of the level of consciousness. Clinicians should counsel families that there is uncertainty regarding the degree of pain and suffering that may be experienced by patients with PDOC (Level B)”. This recommendation stems from conflicting reports in the literature that do not suggest or do not clearly point out pain perception in PDOC [12,88,89,90,91,92,93,94,95], in contrast to the studies that pointed out a conscious pain perception in PDOC [11,36,96]. Consequently, the instrumental assessment of awareness in patients with PDOC is mandatory, because the decisions concerning patients’ preserved level of consciousness in current clinical practice are based principally on clinical observations, which may be fallacious in about 40% of cases [97].

The studies we reviewed suggest that some electrophysiological tools (including LEP, EEG, and ANS functioning) (Figure 1) [98,99,100,101,102,103] may be relevant for pain assessment in PDOC owing also to their remarkable easiness and minimally invasive recording. The available studies lead to two main findings: firstly, all PDOC individuals should be able to perceive primary pain aspects even if only unconsciously; secondly, the presence of complex stimuli elaboration in a way resembling the control subjects and even in the absence of behavioral evidence points out covert awareness in some PDOC individuals, particularly regarding those who have been labeled as with UWS. In this regard, there could be signs of possible covert capacity for subjective pain perception in such patients. These findings might indicate that some patients behaviorally diagnosed as with UWS actually may have a residual capacity of pain perception and are thus in MCS, in line with the theoretical model of cognitive–motor dissociation syndrome [7]. Such complex processing may reflect thalamic and limbic circuits preservation [11,36,101], which are fundamental to awareness recovery following severe brain injury. In addition, the suitability of ANS–CNS functional interaction as possible independent indicators for clinical assessment, diagnosis, and prognosis, and in detecting residual (covert) brain function in PDOC has been documented [65,89,102,103,104,105,106]. Lastly, the electrophysiological measures at rest and in response to stimuli have higher time resolution than functional neuroimaging, can detect rapid changes with variability which is more significant during the day, and are thus proposed as better suitable to capture the complexity of brain dynamics interactions [15,16,105,106,107,108,109,110].

However, the potential of these electrophysiological techniques in detecting pain perception is limited because no established veridical benchmark of level of consciousness or pain perception exists concerning patients with PDOC [51,111,112]. The striking problem is that the assessment of sensitivity and specificity of such approaches regarding residual pain perception evaluation in severely brain-injured patients lacks a checked, clear-cut correspondence between the obtained results and the subjective pain experience [113,114,115,116,117]. There is indeed no established gold standard measure of consciousness and subjective pain perception, beyond the clinical assessment [27,45,51]. Therefore, one paradigm may detect awareness in patients labeled with UWS [118], while another may fail to identify it [119]. This limited sensitivity may depend on specific methodological and analytical aspects of the experimental procedure (including sensitivity and specificity) [51] and by the possibility of deviation of a patient from standard diagnostic categories [17,120,121], particularly regarding pain management [11,33,36,55,56,57,97,122,123,124,125]. Consequently, we will need to know the estimated sensitivity and specificity of these methods, and which stimuli are the most potent in detecting pain responses to implement clinical use of neurophysiological techniques in combination with behavioral assessment of pain.

Furthermore, the evidence regarding the spectral and oscillatory components of the EEG measures and CNS–ANS interactions to detect residual pain awareness is still not consistent, owing to non-homogeneous samples, methodological differences, and data processing [37]. Additionally, the use of such measures to specifically explore pain perception of PDOC patients appears non-systematic. For instance, the intensity and relevance of noxious stimuli may not be sufficient for a patient to detect. Therefore, negative findings must be interpreted with extreme caution; in other words, pain perception can be confirmed empirically, whereas its absence cannot be necessarily demonstrated. Furthermore, CNS and ANS activity can vary over time spontaneously or due to homeostatic or allostatic requirements with different timing and latencies, thus challenging online and single assessments. In addition, the possibility of referencing PDOC data to those coming from healthy controls [126,127,128,129] may not be sufficient, because there could be macroscopic differences in the magnitude of the orienting response (i.e., the automatic reactions to environmental modification and stimuli) [71,130], habituation with repeated stimulations [131,132], and fluctuation in attention and implicit memory [71,133].

One could concern that the detection of purposefully cognitive responses using active paradigms [118,134,135,136,137] is undoubtedly a pillar of PDOC diagnosis [138]. However, there is also some evidence that passive paradigms may be useful for PDOC diagnosis. An in-sequence evaluation of patients using passive stimulation paradigms (e.g., acoustic) may help track awareness recovery [139,140,141]. Therefore, a retrospective analysis of the acquired data may also help prognosticate awareness recovery, but this issue deserves further confirmation. Additionally, assessing the functional integrity of large-scale brain networks may be crucial, because it represents a fundamental prerequisite for awareness to emerge. 

Finally, the instrumental assessment can only offer a quantitative assessment of whether a patient may experience pain. Indeed, qualitative assessment is not achievable. Further research should investigate autonomic and cortical activations expressing covert measures of somatosensory and nociceptive information processing in PDOC patients, preferably using a standardized multi-methods approach.

### 4.1. Management Perspective

The need for a more objective assessment of PDOC patients goes far beyond the differential diagnosis and prognosis because it has relevant consequences on personalizing the rehabilitation approach. In this regard, distinguishing between studies that show neurophysiological pain signatures in patients with PDOC at the individual level beyond a group level is crucial if one aims for the clinical utility of these methods. The studies we reviewed did not identify significant within-group differences concerning neurophysiological pain signatures, with very few exceptions concerning the same patients tested using different, advanced methodological approaches (including GBO, C-LEP spectra, and pain-motor integration) [73,74,75]. Therefore, the electrophysiological test’s diagnostic accuracy, i.e., its clinical utility, derived from calculating the test’s sensitivity and specificity of electrophysiological techniques, is still far from being achieved. This depends on the fact that a correspondence between the test results and the true state-of-affairs is missing because independent methods (i.e., a veridical benchmark of the state-of-affairs or gold standard) checking such correspondence are lacking.

Consequently, even a methodologically accurate electrophysiological paradigm may still be clinically inaccurate, because there are no independent methods for confirmation. However, the clinical utility of electrophysiological assessment benefits from comparing findings in PDOC with the performance of healthy controls under identical experimental conditions (with the abovementioned precautions) [126,127,128,129,130,131,132,133] to increase test sensitivity, thus allowing the use of group-level data in classification models tailored for individual-level analysis, and to enhance specificity (i.e., to obtain as few false-positives as possible) by checking statistical thresholding, and performing multiple independent tests. Further expanding these lines of work, neurophysiological assessment could provide clinicians with useful information concerning conventional (including pharmacological agents and intrathecal baclofen) [142,143,144,145,146] and advanced therapies, such as deep brain stimulation [147,148] and non-invasive neuromodulation (including transcranial current stimulation and rTMS) [149,150]. In this regard, neurophysiological approaches can identify the brain pathways that could be harnessed to foster awareness recovery. However, it is still difficult to differentiate between nociceptive responses and authentic subjective pain experience in understanding pain perception in patients with PDOC. We can only admit that neurophysiological techniques may complement the understanding of pain responses by investigating pain signatures through neurophysiological methods. Indeed, it is surely debated that detecting nociceptive responses implies consciousness, because it does not necessarily verify subjective perceptions of pain, consistently with the limitations as mentioned above on the sensitivity and specificity of the currently available electrophysiological tests. 

Furthermore, as research develops, it will be possible to benefit from neurophysiological data concerning the effectiveness of a treatment strategy [45,151,152]. Lastly, neurophysiological strategies can be implemented to allow patients to communicate (as a brain–computer interface).

### 4.2. Future Research

Solving the bias results toward false positives or false negatives between analysis methods remains a highly challenging obstacle in the research setting. Overcoming the obstacle of false positives or false negatives may consist of reanalyzing previous neuroimaging findings with alternative statistical methods [120,153], analyses [154], comparing different approaches [118,155,156], and studying the robustness of different stimuli [53]. Assuming that negative results during active paradigms do not necessarily imply the absence of cognitive processes, another approach may suggest that a patient with PDOC may process nociceptive stimuli in a way resembling control individuals, thus proposing that they may perceive pain. Alternatively, it is plausible that this patient is misdiagnosed due to the technical constraints of bedside evaluations. 

Of note, research should be oriented at a single-level rather than only at a group-level, if the neurophysiological data have to be adopted as a tool for patient-tailored care plans. The development of technology and methods should aim to increase the availability of objective electrophysiological assessments of functional connectivity and analysis at the level of individual cases and group comparisons [45]. In this regard, a systematic review should deepen the scientific question of the estimated sensitivity and specificity of neurophysiological methods in detecting pain perception in patients with PDOC, particularly the individual-level assessment.

Additionally, larger sample studies are also necessary to enable comparisons among different groups of patients. Lastly, a serial and multimodal neurophysiological assessment may provide us with more profound knowledge of the effects of patients’ clinical features and comorbidity profiles on pain processing to understand better pain perception.

### 4.3. Conclusions

Consistently with the limitations of the clinical assessment, clinicians should be cautious in making definitive conclusions about pain and suffering in individuals with PDOC. The available data suggest that neurophysiology may contribute significantly and actively to clarify the neural correlates of pain perception in patients with PDOCs. Indeed, large-scale brain activations in response to nociceptive stimuli may suggest a conscious perception of pain despite behavioral unresponsiveness. On the other hand, it should be at least assumed that negative results during active paradigms do not necessarily imply the absence of cognitive processes, in keeping with the diagnostic limitations of the electrophysiological paradigms we reviewed. In such cases, further clinical and multiple instrumental evaluations are necessary to prove pain (non)perception [113,114,115,118,155,156,157,158]. In this regard, an accurate clinical and multiple instrumental pain assessment of individuals with PDOC using repeated evaluation through ad hoc clinical scales and neurophysiological approaches may represent the most methodologically sound and ethically responsible approach to avoid misinterpretations and misdiagnoses.

## Figures and Tables

**Figure 1 brainsci-11-00665-f001:**
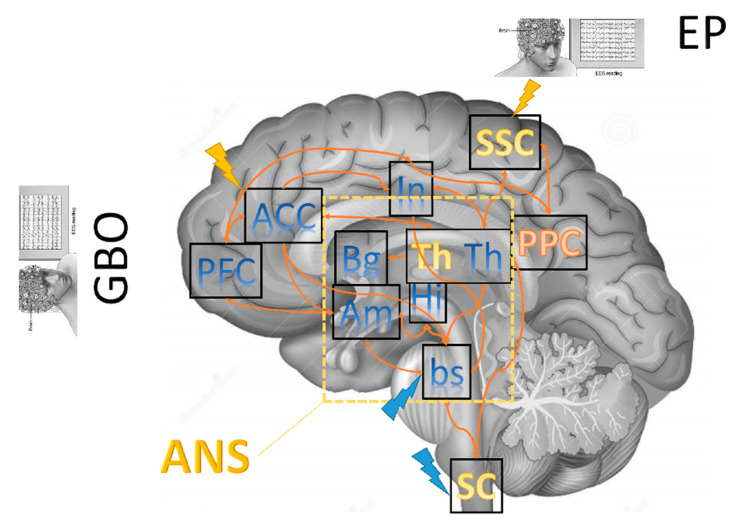
Illustration of the key brain areas related to pain processing and perception. They comprise densely interconnected sensory or discriminative (yellow tags), affective (blue tags), and associative areas (red tags). Brainstem and diencephalic areas are embedded within the autonomic nervous system (ANS) and can be evaluated using specific measures (including hearth rate variability). Sensory and/or nociceptive evoked potentials (EPs) and gamma-band oscillations (GBO) recorded using EEG can follow nociceptive stimulation (blue shock) combined or not with cortical stimulation using, e.g., transcranial magnetic stimulation (yellow shock). These represent the main neurophysiological paradigms available in the literature to gain objective measures of pain processing in PDOC patients. Legend: PFC, prefrontal cortex; ACC, anterior cingulate cortex; In, insula; Bg, basal ganglia; Am, amygdala; Hi, hippocampus; Th, thalamus; bs, brainstem; SC, spinal cord; SSC, somatosensory cortex; PPC, posterior parietal cortex.

**Figure 2 brainsci-11-00665-f002:**
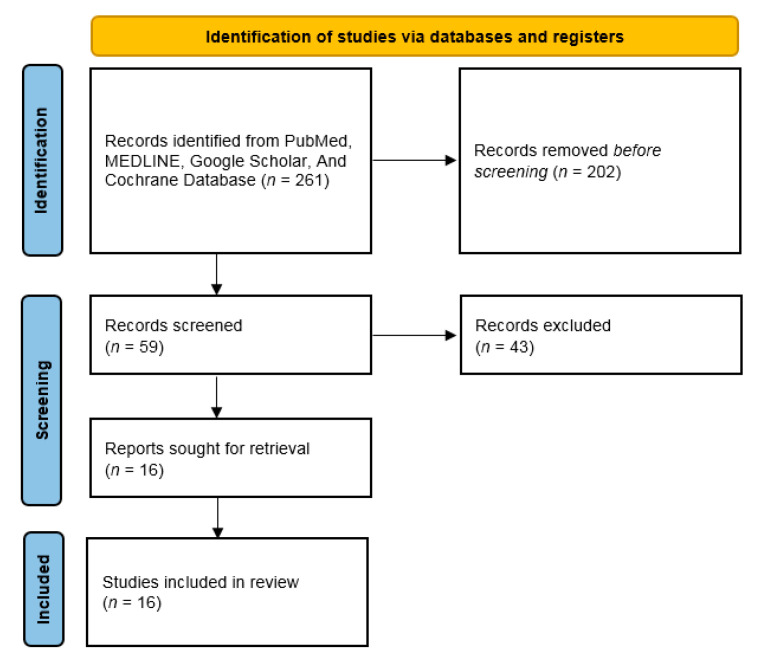
Shows the search strategy we used to select pain assessment studies in patients with PDOC.

**Table 1 brainsci-11-00665-t001:** Main neurophysiological studies investigating pain perception in patients with disorders of consciousness.

Authors	Sample	Methods	Findings	Conclusions
Autonomic Nervous System				
Leo et al., 2016 [59]	12 MCS 10 UWS	GBO of C-LEPs and SR ANS parameters either during a 24-P or following RLS	Only MCS and 2 UWS individuals showed physiological modification of O2 saturation, GBO of C-LEPs and SR either during a 24-P or following RLS	Large-scale ANS parameters and cortical features of advanced pain processing support DOC differential diagnosis and allow identifying residual aware ANS-related cognitive processes
Devalle et al., 2018 [69]	14 UWS 6 MCS	Short-term (<20 s) and long-term (between 20 s and 50 s from noxious stimulus) HRV	Short-term responses in both groups Long-term responses only in MCS	HRV responsiveness differentiates between MCS and UWS
Riganello et al., 2019 [70]	11 MCS 11 UWS 14 HC	HRV assessment using short-term CI	Higher CI in HC compared to DOC at baseline and after noxious stimulation Higher values in MCS versus UWS after noxious stimulation Lower values in noxious versus non-noxious condition in UWS group	UWS have a less complex ANS response to noxious stimuli
Luauté et al., 2018 [72]	7 UWS 6 MCS 7 HC	SCL with stimulations in auditory and olfactory modalities	No different responses in DOC	No DOC distinction
Riganello et al., 2015 [16]	8 UWS	HRV spectrum	Significant correlation between HRV spectral features and CRS-R	The timely variability of ANS tone serves as an indicator for diagnosis and prognosis
Venturella et al., 2019 [79]	21 UWS	ANS responsiveness to touch- and pain-related stimuli	Fronto-parietal activation in both modalities. Increase in delta oscillations, electrodermal activity, and HRV following painful stimuli	Stimuli can capture basic attention orientation and perceptual processes. Only nociceptive stimulation seems entraining cognitive processes at an aware level
Laser-Evoked Potentials and Advanced EEG Signal Analyses				
De Tommaso et al., 2013 [75]	3 UWS 4 MCS 11HC	LEP SEPs AMN	LEPs in all patients Significant N2 and P2 latency increase No SEPs in all patients but one MCS AMN in all patients	Possible pain processing preservation despite sensory impairment
De Tommaso et al., 2015 [76]	5 UWS 4 MCS 11 HC	LEP multimodal EP	Constant preservation of LEP despite a variable degree of preservation of the other EPs	Possible pain processing preservation despite sensory impairment
De Salvo et al., 2015 [78]	13 UWS 10 MCS	LEP	Lower amplitudes and more delayed in UWS than MCS	LEP features can discriminate between MCS and UWS
Naro et al., 2015 [79]	23 UWS 15 MCS 15 HC	Aδ-LEP C-LEP	Higher amplitudes and less delayed latencies in HC than DOC Higher amplitudes and less delayed latencies in MCS than UWS Some UWS showed only C-LEP	The residual presence of C-LEP should be assessed when Aδ-LEP are missing, because a potential pain experience should be still present in some patients
Naro et al., 2015 [80]	10 UWS 10 HC	MEP LEP PMI	PMI deterioration in DOC, more in UWS than MCS PMI preserved in some UWS	Residual plasticity properties at large-scale cortical level suggesting residual pain awareness
Naro et al., 2016 [81]	18 UWS 15 MCS	GBO following RLS	Increase in GBO power and NCS-R score in HC, MCS and 5 UWS	Presence of aware pain processing as per GBO modulation
Aricò et al., 2016 [82]	8 UWS 6 MCS	LEP 24 h polysomnography	Higher LEP latencies and lower amplitudes in UWS than MCS Spared sleep structure in MCS compared to UWS in correlation with LEP findings	Preserved sleep structure and pain processing require a spared global brain connectivity, which expresses thalamo–cortical functionality supporting consciousness
Naro et al., 2017 [83]	10 UWS 10 MCS 10 HC	IPI variability of LEP components	Correlation between IPI and NCS-R	IPI variability might represent an objective measure of pain processing
Calabrò et al., 2017 [84]	11 UWS 10 MCS	γ-band LORETA activations, GBO, and HRV following RLS	Spared γ-band LORETA activations, GBO, and HRV in MCS and two UWS (with brain activation limited to limbic areas)	Nearly physiologic pain processing in MCS; connectivity breakdown in UWS, which limits aware pain perception to residual
Naro et al., 2015 [85]	10 UWS 10 MCS 10 HC	1 Hz rTMS over ACC affecting frontal GBO and EEP	Increase in GBO and decrease in EPP in MCS and two UWS subjects Decreased pain rating in HC (as per VAS) and MCS (as per NCS-R)	ACC rTMS aftereffects suggest aware pain processing

## Data Availability

Not applicable.

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
