# Peer review of "Pain Perception in Disorder of Consciousness: A Scoping Review on Current Knowledge, Clinical Applications, and Future Perspective"

_brainsci, 2021, doi:10.3390/brainsci11050665_

Round 1
Reviewer 1 Report
This is an interesting review on pain perception in disorder of consciousness, while my main concern is the literature searching protocol. Since the number of publications on this topic is very limited, it is therefore critical to follow the standard protocol for such a review.
Comments:
- the searching protocol needs to be more specified so as to show its objective and reproductive. For example, did the authors limit the range of published years? Was the screening of the papers performed by at least two independent experts? How did they rank the papers? A flow chart would be recommended to show the whole process.
- it would be better to spell out the abbreviations (SSC, PPC, ect.) in Figure 1, or simply remove them if not relevant to the review.
- 'PDOC' and 'pDOC' are used alternatively, it is necessary to keep consistent throughout this manuscript
- please double check the format: e.g. 1) there is no space between the order number and 1st author for the first nine refs, and 2) content is missing for Ref. 114-116
Author Response
This is an interesting review on pain perception in disorder of consciousness, while my main concern is the literature searching protocol. Since the number of publications on this topic is very limited, it is therefore critical to follow the standard protocol for such a review.
We thank the reviewer for the appreciation to our ms and the useful suggestions to improve its quality. As suggested, we better detailed the section research strategy to describe how the papers were selected for the study, on what criteria, and methodology. A flow-chart was added as recommended.
Comments:
- the searching protocol needs to be more specified so as to show its objective and reproductive. For example, did the authors limit the range of published years? Was the screening of the papers performed by at least two independent experts? How did they rank the papers? A flow chart would be recommended to show the whole process.
As suggested, we better detailed the section research strategy to describe how the papers were selected for the study, on what criteria, and methodology. A flow-chart was added as recommended.
- it would be better to spell out the abbreviations (SSC, PPC, ect.) in Figure 1, or simply remove them if not relevant to the review.
Done.
- 'PDOC' and 'pDOC' are used alternatively, it is necessary to keep consistent throughout this manuscript
Done.
- please double check the format: e.g. 1) there is no space between the order number and 1st author for the first nine refs, and 2) content is missing for Ref. 114-116
Amended.
Kindest regards,
The authors
Reviewer 2 Report
Reply to authors of the manuscript “Pain Perception in Disorder of Consciousness: Current Knowledge, Clinical Applications, and Future Perspective”
The topic of this manuscript is very welcome, and I merit the authors for putting effort into the scientific review of the increased knowledge with regard to the clinical value of electrophysiological techniques in assessing and detecting signs of pain awareness in patients with disorders of consciousness (DoC). Thus, the manuscript contributes potentially with important pooled knowledge of pain perception and neuroscientific assessment techniques to supplement clinical assessment of pain in this patient group.
However, I have some major concerns about the manuscript in its present form.
One major concern about the manuscript is that it lacks precision and details. For example, please give a precise definition of the diagnostic criteria for UWS (also referred to as vegetative state) and MCS, as the entities are insufficiently described in the manuscript in its present form. Herein, original references to the diagnostic criteria should be included.
Also, there are several odd English phrases and sentences, and the manuscript is in need of thorough language editing. The list of references is not complete, see ref 114-116.
It needs to be clarified whether the manuscript is a scoping review or a systematic review. See recommended guidelines: https://www.equator-network.org
There are specific methods for how to conduct a review, and to ensure transparency of the review process of included studies. These are the established PRISMA guidelines. For instance, it is a major shortcoming in the presented manuscript that inclusions and exclusion criteria of the included studies are not specified. It should definitely be an inclusion criterion that eligible studies have applied a sound assessment tool for the behavioral assessment of the level of consciousness in the study patients. Tools with acceptable psychometric properties have been highlighted in the systematic review of Seel et al., 2010, eg. the CRS-R.
Authors should tone down the certainty of being able to detect covert signs of consciousness by LEP or similar neurophysiological techniques. The authors do not incorporate in their interpretation and argumentation for the potential of these techniques in detecting pain perception or conscious tactile perception, the fact that no established veridical benchmark of level of consciousness exists with regard to patients with DoC. This is the core problem in all scientific studies that investigate the usefulness of modern neuroscientific methods in detecting signs of consciousness, including subjective pain perception, in patients with DoC. True estimates of sensitivity and specificity of electrophysiological techniques designed to detect residual consciousness/ pain perception in severely brain-injured patients is problematic due to the lack of independent methods for checking the correspondence of test results with the true level of consciousness or the true subjective pain experience in these patients, i.e. there is no established gold standard measure of consciousness/ subjective pain perception.
Authors do not distinguish between studies that have been able to show neurophysiological “pain signature” in patients with pDoC on a group level and studies that investigate pain responses with neurophysiological techniques in patients with pDoC at an individual level. This is crucial if the manuscript aims at reasoning for clinical utility of these methods.
Also, the authors need to incorporate that there is a lack of clear cut-off in the neurophysiological responses for when a patient can be verified as UWS, MCS, or verified as a patient in pDoC with subjective pain perception in contrast to only eliciting nociceptive responses on an individual level. Studies showing pain responses in DoC at a group level have limited value when trying to understand pain perception at an individual level. The authors already point out that many of the neurophysiological “brain signatures” are features that emerge on a continuum, in other word there is no clear cut-off for distinguishing conscious perception in these patients with the use of electrophysiological techniques. Within the area of understanding pain perception in patients with pDoC, it is therefore still difficult to differentiate between nociceptive responses and true subjective pain experience. Detecting nociceptive responses in patients with DOC does necessarily infer subjective pain perception, but neurophysiological techniques may complement the understanding of pain responses by investigating “pain signatures” by means of neurophysiological methods. However, in a clinical setting, one need to know the sensitivity and specificity of these techniques in detecting subjective pain perception (at an individual level). The manuscript does not systematically investigate these questions, and therefore this perspective should be taken into account in the description of the results of the included studies, not only as a final remark in the discussion. This is essential also with regard to the recommendation the authors bring forth in the discussion. To implement clinical use of neurophysiological techniques in combination with behavioral assessment of pain, one need to know the estimated sensitivity and specificity of these methods, as well as which stimuli is the most potent in detecting pain responses. The authors describe the difference between nociception and pain perception in the discussion, but these two terminologies should be introduced and described in the introduction.
Based on these comments, I would recommend the authors to define this manuscript as a scoping review. Further, I would encourage the authors as a next phase to do a systematic review and go into the scientific question of the estimated sensitivity and specificity of the neurophysiological methods in detecting pain perception in patients with DoC. The would be warranted:-)
The authors interpret findings in some studies phrased as preserved pain perception in patients in UWS. This contradicts the diagnostic criteria of the entity. (see line 206/ line 232). The core definition of UWS is lack of consciousness. Would it not be more correct to describe this in terms of signs of possible covert capacity for subjective pain perception, and that such findings might indicate that some patients behaviorally diagnosed as in UWS actually may have residual capacity of pain perception, and are thus in MCS? The latter interpretation is also in line with the theoretical model of Schiff et al. called the “cognitive motor dissociation syndrome”, referred in the introduction of the manuscript. See for instance how line 63/64 is phrased. The case described in line 243 might also be an example of a patient with CMD.
Author Response
The topic of this manuscript is very welcome, and I merit the authors for putting effort into the scientific review of the increased knowledge with regard to the clinical value of electrophysiological techniques in assessing and detecting signs of pain awareness in patients with disorders of consciousness (DoC). Thus, the manuscript contributes potentially with important pooled knowledge of pain perception and neuroscientific assessment techniques to supplement clinical assessment of pain in this patient group. However, I have some major concerns about the manuscript in its present form.
We thank the reviewer for the appreciation to our ms and the useful suggestions to improve its quality.
- One major concern about the manuscript is that it lacks precision and details. For example, please give a precise definition of the diagnostic criteria for UWS (also referred to as vegetative state) and MCS, as the entities are insufficiently described in the manuscript in its present form. Herein, original references to the diagnostic criteria should be included.
We added the missing information, as correctly suggested.
- Also, there are several odd English phrases and sentences, and the manuscript is in need of thorough language editing.
English language was revised by a professional native English speaker.
- The list of references is not complete, see ref 114-116.
Amended.
- It needs to be clarified whether the manuscript is a scoping review or a systematic review. See recommended guidelines: https://www.equator-network.org There are specific methods for how to conduct a review, and to ensure transparency of the review process of included studies. These are the established PRISMA guidelines. For instance, it is a major shortcoming in the presented manuscript that inclusions and exclusion criteria of the included studies are not specified. It should definitely be an inclusion criterion that eligible studies have applied a sound assessment tool for the behavioral assessment of the level of consciousness in the study patients. Tools with acceptable psychometric properties have been highlighted in the systematic review of Seel et al., 2010, eg. the CRS-R.
As suggested, we better detailed the section research strategy of this scoping review to describe how the papers were selected for the study, on what criteria, and methodology. Particularly, we specified that the present work was a scoping review as this was aimed at describing the landscape of available evidence, identifying gaps in the existing literature, and illuminating areas for further research. A scoping review methodology was also selected because of the relatively specific nature of the field. In addition, we specified the inclusion and exclusion criteria (sec. 2 - research strategy). As correctly suggested, it was ascertained that the study patients were provided with pain-related behavioral assessment using tools with acceptable psychometric properties, including NCS and CRS-R.
- Authors should tone down the certainty of being able to detect covert signs of consciousness by LEP or similar neurophysiological techniques. The authors do not incorporate in their interpretation and argumentation for the potential of these techniques in detecting pain perception or conscious tactile perception, the fact that no established veridical benchmark of level of consciousness exists with regard to patients with DoC. This is the core problem in all scientific studies that investigate the usefulness of modern neuroscientific methods in detecting signs of consciousness, including subjective pain perception, in patients with DoC. True estimates of sensitivity and specificity of electrophysiological techniques designed to detect residual consciousness/ pain perception in severely brain-injured patients is problematic due to the lack of independent methods for checking the correspondence of test results with the true level of consciousness or the true subjective pain experience in these patients, i.e. there is no established gold standard measure of consciousness/ subjective pain perception.
We thank the reviewer for this interesting food for thought, which was elaborated and discussed (see p. 11).
- Authors do not distinguish between studies that have been able to show neurophysiological “pain signature” in patients with pDoC on a group level and studies that investigate pain responses with neurophysiological techniques in patients with pDoC at an individual level. This is crucial if the manuscript aims at reasoning for clinical utility of these methods. Also, the authors need to incorporate that there is a lack of clear cut-off in the neurophysiological responses for when a patient can be verified as UWS, MCS, or verified as a patient in pDoC with subjective pain perception in contrast to only eliciting nociceptive responses on an individual level. Studies showing pain responses in DoC at a group level have limited value when trying to understand pain perception at an individual level. The authors already point out that many of the neurophysiological “brain signatures” are features that emerge on a continuum, in other word there is no clear cut-off for distinguishing conscious perception in these patients with the use of electrophysiological techniques. Within the area of understanding pain perception in patients with pDoC, it is therefore still difficult to differentiate between nociceptive responses and true subjective pain experience. Detecting nociceptive responses in patients with DOC does necessarily infer subjective pain perception, but neurophysiological techniques may complement the understanding of pain responses by investigating “pain signatures” by means of neurophysiological methods. However, in a clinical setting, one need to know the sensitivity and specificity of these techniques in detecting subjective pain perception (at an individual level). The manuscript does not systematically investigate these questions, and therefore this perspective should be taken into account in the description of the results of the included studies, not only as a final remark in the discussion. This is essential also with regard to the recommendation the authors bring forth in the discussion. To implement clinical use of neurophysiological techniques in combination with behavioral assessment of pain, one need to know the estimated sensitivity and specificity of these methods, as well as which stimuli is the most potent in detecting pain responses.
We thank the reviewer for this useful suggestion. We better highlighted whether the reviewed studies provided within-group beyond between-group differentiation of patients with PDOC in the result section. We also expanded the issue that research should be oriented at single-level rather than only group level, if the neuro-physiological data have to be adopted as a tool for patient-tailored care plans. Actually, the development of technology and methods should aim to increase the availability of objective electrophysiological assessment of functional connectivity and analysis at the level of individual cases as well as group comparisons (see p. 12).
- The authors describe the difference between nociception and pain perception in the discussion, but these two terminologies should be introduced and described in the introduction.
Done.
- Based on these comments, I would recommend the authors to define this manuscript as a scoping review. Further, I would encourage the authors as a next phase to do a systematic review and go into the scientific question of the estimated sensitivity and specificity of the neurophysiological methods in detecting pain perception in patients with DoC. The would be warranted:-)
We totally agree with reviewer’s point of view. We appreciated very much his/her argumentations and suggestions, which were helpful to improve the quality of our work. We also capitalized from reviewer’s suggestion for the next work.
- The authors interpret findings in some studies phrased as preserved pain perception in patients in UWS. This contradicts the diagnostic criteria of the entity. (see line 206/ line 232). The core definition of UWS is lack of consciousness. Would it not be more correct to describe this in terms of signs of possible covert capacity for subjective pain perception, and that such findings might indicate that some patients behaviorally diagnosed as in UWS actually may have residual capacity of pain perception, and are thus in MCS? The latter interpretation is also in line with the theoretical model of Schiff et al. called the “cognitive motor dissociation syndrome”, referred in the introduction of the manuscript. See for instance how line 63/64 is phrased. The case described in line 243 might also be an example of a patient with CMD.
We agree on reviewer’s opinion and the text was amended consistently.
Kindest regards,
The authors.
Round 2
Reviewer 2 Report
I thank the authors for adjusting the manuscript in accord to the recommendations of the first review round.
I am still persistent that the manuscript should follow the check list and guidelines for a scoping review (see Tricco et al, Annals of Internal Medicine, 2018). The guidelines and checklist therefore include the reporting of eligibility criteria, description of information sources (databases), and report of full electronic search. Please report on all checklist reporting items.
Line 140. Not only variation in statistical methodology, but could also stem from variation in robustness stimuli. See for example systematic review of Hauger et al., J Head Trauma Rehabil., 2017 for description various sources that can explain the wide variety with regard to sensitivity and specificity of electrophysiological techniques in detecting consciousness in patients with DoC.
Line 163. write out fully rehabilitation (not rehab).
Line 186. Auhors should differentiate between the inclusion criteria that state that all eligible studies should include a behavioral assessment tool for assessing the level of consciousness in the included subjects (eg. CRS-R) and refer to the tools that have acceptable psychometric properties in accord with the thorough systematic review of Seel et al. 2010. The NCS-R is on the other hand a behavioral tool for detecting pain in patients with DoC. This should be distinguished.
Also, the authors have defined their scoping review to include patients with PDOC. Have they ensured an inclusion criterion of eligible studies that includes patients with DOC> 28 days?
Table 1 looks unfinished. Under column 2, the data have bullet points some places and other places not. Same for column 3, data summarized for a study should start with a capital letter, for readers to better lead the table. Please be accurate.
Line 195. This paragraph is confusing. The authors list various physiological variables such as eye- opening, breathing, heart rate, blood pressure. Firstly, is eye-opening a physiological variable? Secondly, are these variables proof of pain perception? I believe not. For instance, eye-opening can be mere spontaneous behavioral responses. Please rephrase (objective measures to prove…)
Line 210. I would be careful with using words as “proof”
Line 324. Replace the word “unwarily” with “unconsciously”
The conclusion section is a bit lengthy, and could possibly benefit from more focused and condensed phrasing.
Line 419. “…..detecting nociceptive responses in patients with DOC does necessarily infer subjective pain perception”. It is surely debated that detecting nociceptive responses infer consciousness, as it not necessarily verifies subjective perception of pain.
Line 432. -And also studying the robustness of different stimuli.
Author Response
I thank the authors for adjusting the manuscript in accord to the recommendations of the first review round.
We thank the reviewer for the appreciation to our ms and the useful suggestions to improve its quality further.
- I am still persistent that the manuscript should follow the check list and guidelines for a scoping review (see Tricco et al, Annals of Internal Medicine, 2018). The guidelines and checklist therefore include the reporting of eligibility criteria, description of information sources (databases), and report of full electronic search. Please report on all checklist reporting items.
We were more careful in following the checklist and guidelines for a scoping review, as recommended by the reviewer. We added more details on the entire research strategy, consistently with PRISMA-ScR.
- Line 140. Not only variation in statistical methodology, but could also stem from variation in robustness stimuli. See for example systematic review of Hauger et al., J Head Trauma Rehabil., 2017 for description various sources that can explain the wide variety with regard to sensitivity and specificity of electrophysiological techniques in detecting consciousness in patients with DoC.
We thank the reviewer for this information that was incorporated in the text.
- Line 163. write out fully rehabilitation(not rehab).
Amended
- Line 186. Auhors should differentiate between the inclusion criteria that state that all eligible studies should include a behavioral assessment tool for assessing the level of consciousness in the included subjects (eg. CRS-R) and refer to the tools that have acceptable psychometric properties in accord with the thorough systematic review of Seel et al. 2010. The NCS-R is on the other hand a behavioral tool for detecting pain in patients with DoC. This should be distinguished.
We agree on reviewer’s suggestion. The sentence was revised accordingly, since misleading.
- Also, the authors have defined their scoping review to include patients with PDOC. Have they ensured an inclusion criterion of eligible studies that includes patients with DOC> 28 days?
We now better specified that we ensured an inclusion criterion of eligible studies that includes patients with DOC> 28 days
- Table 1 looks unfinished. Under column 2, the data have bullet points some places and other places not. Same for column 3, data summarized for a study should start with a capital letter, for readers to better lead the table. Please be accurate.
Checked and corrected.
- Line 195. This paragraph is confusing. The authors list various physiological variables such as eye- opening, breathing, heart rate, blood pressure. Firstly, is eye-opening a physiological variable? Secondly, are these variables proof of pain perception? I believe not. For instance, eye-opening can be mere spontaneous behavioral responses. Please rephrase (objective measures to prove…)
The entire paragraph was reworded as misleading, as correctly pointed out by the reviewer. We better specified that we generally referred to a three-order measures of pain processing, i.e., the spontaneous responses or behaviors concerning pain, those that are elicited using nociceptive stimuli, or that are elicited by active stimulation paradigms.
- Line 210. I would be careful with using words as “proof”
Amended.
- Line 324. Replace the word “unwarily” with “unconsciously”
Corrected.
- The conclusion section is a bit lengthy, and could possibly benefit from more focused and condensed phrasing.
The section was revised to make it more concise and focused on.
- Line 419. “…..detecting nociceptive responses in patients with DOC does necessarily infer subjective pain perception”. It is surely debated that detecting nociceptive responses infer consciousness, as it not necessarily verifies subjective perception of pain.
Amended as per reviewer suggestion.
- Line 432. -And also studying the robustness of different stimuli.
Added.
Kindest regards,
The authors